# LEARNING APPROXIMATE INFERENCE NETWORKS FOR STRUCTURED PREDICTION

**Lifu Tu**    **Kevin Gimpel**
Toyota Technological Institute at Chicago, Chicago, IL, 60637, USA
`{lifu,kgimpel}@ttic.edu`

## ABSTRACT

Structured prediction energy networks (SPENs; Belanger & McCallum 2016) use neural network architectures to define energy functions that can capture arbitrary dependencies among parts of structured outputs. Prior work used gradient descent for inference, relaxing the structured output to a set of continuous variables and then optimizing the energy with respect to them. We replace this use of gradient descent with a neural network trained to approximate structured $\mathrm{argmax}$ inference. This "inference network" outputs continuous values that we treat as the output structure. We develop large-margin training criteria for joint training of the structured energy function and inference network. On multi-label classification we report speed-ups of 10-60x compared to (Belanger et al., 2017) while also improving accuracy. For sequence labeling with simple structured energies, our approach performs comparably to exact inference while being much faster at test time. We then demonstrate improved accuracy by augmenting the energy with a "label language model" that scores entire output label sequences, showing it can improve handling of long-distance dependencies in part-of-speech tagging. Finally, we show how inference networks can replace dynamic programming for test-time inference in conditional random fields, suggestive for their general use for fast inference in structured settings.

## 1 INTRODUCTION

Energy-based modeling (LeCun et al., 2006) associates a scalar measure of compatibility to each configuration of input and output variables. Given an input $x$, the predicted output $\hat{y}$ is chosen by minimizing an **energy function** $E(x, \hat{y})$. For structured prediction, the parameterization of the energy function can leverage domain knowledge about the structured output space. However, learning and prediction become complex.

Structured prediction energy networks (SPENs; Belanger & McCallum 2016) use an energy function to score structured outputs, and perform inference by using gradient descent to iteratively optimize the energy with respect to the outputs. Belanger et al. (2017) develop an "end-to-end" method that unrolls an approximate energy minimization algorithm into a fixed-size computation graph that is trainable by gradient descent. After learning the energy function, however, they still must use gradient descent for test-time inference.

We replace the gradient descent approach with a neural network trained to do inference, which we call an **inference network**. It can have any architecture such that it takes an input $x$ and returns an output interpretable as a $y$. As in prior work, we relax $y$ from discrete to continuous. For multi-label classification, we use a feed-forward network that outputs a vector. We assign a single label to each dimension of the vector, interpreting its value as the probability of predicting that label. For sequence labeling, we output a distribution over predicted labels at each position in the sequence. We adapt the energy functions such that they can operate with both discrete ground truth outputs and outputs generated by our inference networks.

We define large-margin training objectives to jointly train energy functions and inference networks. Our training objectives resemble the alternating optimization framework of generative adversarial networks (GANs; Goodfellow et al. 2014): the inference network is analogous to the generator and the energy function is analogous to the discriminator. Our approach avoids $\mathrm{argmax}$ computations,

making training and test-time inference faster than standard SPENs. We experiment with multi-label classification using the same setup as Belanger & McCallum (2016), demonstrating speed-ups of 10x in training time and 60x in test-time inference while also improving accuracy.

We then design a SPEN and inference network for sequence labeling by using recurrent neural networks (RNNs). We perform comparably to a conditional random field (CRF; Lafferty et al. 2001) when using the same energy function, with faster test-time inference. We also experiment with a richer energy that includes a "label language model" that scores entire output label sequences using an RNN, showing it can improve handling of long-distance dependencies in part-of-speech tagging. Finally, we show how inference networks can replace dynamic programming for test-time inference with CRFs, suggestive for the general use of inference networks to speed up inference in traditional structured prediction settings.

## 2 STRUCTURED PREDICTION ENERGY NETWORKS

We denote the space of inputs by $\mathcal{X}$. For a given input $\boldsymbol{x} \in \mathcal{X}$, we denote the space of legal structured outputs by $\mathcal{Y}(\boldsymbol{x})$. We denote the entire space of structured outputs by $\mathcal{Y} = \cup_{\boldsymbol{x} \in \mathcal{X}} \mathcal{Y}(\boldsymbol{x})$. A SPEN defines an **energy function** $E_\Theta : \mathcal{X} \times \mathcal{Y} \to \mathbb{R}$ parameterized by $\Theta$ that uses a functional architecture to compute a scalar energy for an input/output pair.

We describe the SPEN for multi-label classification (MLC) from Belanger & McCallum (2016). Here, $\boldsymbol{x}$ is a fixed-length feature vector. We assume there are $L$ labels, each of which can be on or off for each input, so $\mathcal{Y}(\boldsymbol{x}) = \{0, 1\}^L$ for all $\boldsymbol{x}$. The energy function is the sum of two terms: $E_\Theta(\boldsymbol{x}, \boldsymbol{y}) = E^{loc}(\boldsymbol{x}, \boldsymbol{y}) + E^{lab}(\boldsymbol{y})$. $E^{loc}(\boldsymbol{x}, \boldsymbol{y})$ is the sum of linear models:

$$E^{loc}(\boldsymbol{x}, \boldsymbol{y}) = \sum_{i=1}^{L} y_i b_i^\top F(\boldsymbol{x}) \tag{1}$$

where $b_i$ is a parameter vector for label $i$ and $F(\boldsymbol{x})$ is a multi-layer perceptron computing a feature representation for the input $\boldsymbol{x}$. $E^{lab}(\boldsymbol{y})$ scores $\boldsymbol{y}$ independent of $\boldsymbol{x}$:

$$E^{lab}(\boldsymbol{y}) = c_2^\top g(C_1 \boldsymbol{y}) \tag{2}$$

where $c_2$ is a parameter vector, $g$ is an elementwise non-linearity function, and $C_1$ is a parameter matrix. After learning the energy function, prediction minimizes energy:

$$\hat{\boldsymbol{y}} = \operatorname*{argmin}_{\boldsymbol{y} \in \mathcal{Y}(\boldsymbol{x})} E_\Theta(\boldsymbol{x}, \boldsymbol{y}) \tag{3}$$

However, solving Eq. (3) requires combinatorial algorithms because $\mathcal{Y}$ is a discrete structured space. This becomes intractable when $E_\Theta$ does not decompose into a sum over small "parts" of $\boldsymbol{y}$. Belanger & McCallum (2016) relax this problem by allowing the discrete vector $\boldsymbol{y}$ to be continuous. We use $\mathcal{Y}_R$ to denote the relaxed output space. For MLC, $\mathcal{Y}_R(\boldsymbol{x}) = [0, 1]^L$. They solve the relaxed problem by using gradient descent to iteratively optimize the energy with respect to $\boldsymbol{y}$. Since they train with a structured large-margin objective, repeated inference is required during learning. They note that using gradient descent for this inference step is time-consuming and makes learning less stable. So Belanger et al. (2017) propose an "end-to-end" learning procedure inspired by Domke (2012). This approach performs backpropagation through each step of gradient descent. We compare to both methods in our experiments below.

## 3 INFERENCE NETWORKS FOR SPENS

Belanger & McCallum (2016) relaxed $\boldsymbol{y}$ from a discrete to a continuous vector and used gradient descent for inference. We also relax $\boldsymbol{y}$ but we use a different strategy to approximate inference. We define an **inference network** $\mathbf{A}_\Psi(\boldsymbol{x})$ parameterized by $\Psi$ and train it with the goal that

$$\mathbf{A}_\Psi(\boldsymbol{x}) \approx \operatorname*{argmin}_{\boldsymbol{y} \in \mathcal{Y}_R(\boldsymbol{x})} E_\Theta(\boldsymbol{x}, \boldsymbol{y}) \tag{4}$$

Given an energy function $E_\Theta$ and a dataset $X$ of inputs, we solve the following optimization problem:

$$\hat{\Psi} \leftarrow \operatorname*{argmin}_{\Psi} \sum_{\boldsymbol{x} \in X} E_\Theta(\boldsymbol{x}, \mathbf{A}_\Psi(\boldsymbol{x})) \tag{5}$$

The architecture of $\mathbf{A}_\Psi$ will depend on the task. For MLC, the same set of labels is applicable to every input, so $\boldsymbol{y}$ has the same length for all inputs. So, we can use a feed-forward network for $\mathbf{A}_\Psi$ with a vector output, treating each dimension as the prediction for a single label. For sequence labeling, each $\boldsymbol{x}$ (and therefore each $\boldsymbol{y}$) can have a different length, so we must use a network architecture for $\mathbf{A}_\Psi$ that permits different lengths of predictions. We use an RNN that returns a vector at each position of $\boldsymbol{x}$. We interpret this vector as a probability distribution over output labels at that position.

We note that the output of $\mathbf{A}_\Psi$ must be compatible with the energy function, which is typically defined in terms of the original discrete output space $\mathcal{Y}$. This may require generalizing the energy function to be able to operate both on elements of $\mathcal{Y}$ and $\mathcal{Y}_R$. For MLC, no change is required. For sequence labeling, the change is straightforward and is described below in Section 7.2.1.

## 4 JOINT TRAINING OF SPENS AND INFERENCE NETWORKS

Belanger & McCallum (2016) propose a structured hinge loss for training SPENs:

$$\min_\Theta \sum_{\langle \boldsymbol{x}_i, \boldsymbol{y}_i \rangle \in \mathcal{D}} \left[ \max_{\boldsymbol{y} \in \mathcal{Y}_R(\boldsymbol{x})} \left( \triangle(\boldsymbol{y}, \boldsymbol{y}_i) - E_\Theta(\boldsymbol{x}_i, \boldsymbol{y}) + E_\Theta(\boldsymbol{x}_i, \boldsymbol{y}_i) \right) \right]_+ \tag{6}$$

where $\mathcal{D}$ is the set of training pairs, $[f]_+ = \max(0, f)$, and $\triangle(\boldsymbol{y}, \boldsymbol{y}')$ is a structured **cost** function that returns a nonnegative value indicating the difference between $\boldsymbol{y}$ and $\boldsymbol{y}'$. This loss is often referred to as "margin-rescaled" structured hinge loss (Taskar et al., 2004; Tsochantaridis et al., 2005).

However, this loss is expensive to minimize for structured models because of the "cost-augmented" inference step ($\max_{\boldsymbol{y} \in \mathcal{Y}_R(\boldsymbol{x})}$). In prior work with SPENs, this step used gradient descent. We replace this with a **cost-augmented inference network** $\mathbf{A}_\Phi(\boldsymbol{x})$. As suggested by the notation, the cost-augmented inference network $\mathbf{A}_\Phi$ and the inference network $\mathbf{A}_\Psi$ will typically have the same functional form, but use different parameters $\Phi$ and $\Psi$. We write our new optimization problem as:

$$\min_\Theta \max_\Phi \sum_{\langle \boldsymbol{x}_i, \boldsymbol{y}_i \rangle \in \mathcal{D}} \left[ \triangle(\mathbf{A}_\Phi(\boldsymbol{x}_i), \boldsymbol{y}_i) - E_\Theta(\boldsymbol{x}_i, \mathbf{A}_\Phi(\boldsymbol{x}_i)) + E_\Theta(\boldsymbol{x}_i, \boldsymbol{y}_i) \right]_+ \tag{7}$$

We treat this optimization problem as a minimax game and find a saddle point for the game. Following Goodfellow et al. (2014), we implement this using an iterative numerical approach. We alternatively optimize $\Phi$ and $\Theta$, holding the other fixed. Optimizing $\Phi$ to completion in the inner loop of training is computationally prohibitive and may lead to overfitting. So we alternate between one mini-batch for optimizing $\Phi$ and one for optimizing $\Theta$. We also add $L_2$ regularization terms for $\Theta$ and $\Phi$.

The objective for the cost-augmented inference network is:

$$\hat{\Phi} \leftarrow \operatorname*{argmax}_\Phi [\triangle(\mathbf{A}_\Phi(\boldsymbol{x}_i), \boldsymbol{y}_i) - E_\Theta(\boldsymbol{x}_i, \mathbf{A}_\Phi(\boldsymbol{x})_i) + E_\Theta(\boldsymbol{x}_i, \boldsymbol{y}_i)]_+ \tag{8}$$

That is, we update $\Phi$ so that $\mathbf{A}_\Phi$ yields an output that has low energy and high cost, in order to mimic cost-augmented inference. The energy parameters $\Theta$ are kept fixed. There is an analogy here to the generator in GANs: $\mathbf{A}_\Phi$ is trained to produce a high-cost structured output that is also appealing to the current energy function. To help stabilize training of $\Phi$, we add several terms to this objective, discussed below in Section 5.

The objective for the energy function is:

$$\hat{\Theta} \leftarrow \operatorname*{argmin}_\Theta [\triangle(\mathbf{A}_\Phi(\boldsymbol{x}_i), \boldsymbol{y}_i) - E_\Theta(\boldsymbol{x}_i, \mathbf{A}_\Phi(\boldsymbol{x}_i)) + E_\Theta(\boldsymbol{x}_i, \boldsymbol{y}_i)]_+ + \lambda \|\Theta\|_2^2 \tag{9}$$

That is, we update $\Theta$ so as to widen the gap between the cost-augmented and ground truth outputs. There is an analogy here to the discriminator in GANs. The energy function is updated so as to enable it to distinguish "fake" outputs produced by $\mathbf{A}_\Phi$ from real outputs $\boldsymbol{y}_i$.

Training iterates between updating $\Phi$ and $\Theta$ using the objectives above.

### 4.1 TEST-TIME INFERENCE

After training, we want to use an inference network $\mathbf{A}_\Psi$ defined in Eq. (4). However, training only gives us a cost-augmented inference network $\mathbf{A}_\Phi$. Since $\mathbf{A}_\Psi$ and $\mathbf{A}_\Phi$ have the same functional form,

we can use $\Phi$ to initialize $\Psi$, then do additional training on $\mathbf{A}_\Psi$ as in Eq. (5) where $X$ is the training or validation set. This step helps the resulting inference network to produce outputs with lower energy, as it is no longer affected by the cost function. Since this procedure does not use the output labels of the $\boldsymbol{x}$'s in $X$, it could also be applied to the test data in a transductive setting.

## 4.2 VARIATIONS AND SPECIAL CASES

This approach also permits us to use large-margin structured prediction with slack rescaling (Tsochantaridis et al., 2005). Slack rescaling can yield higher accuracies than margin rescaling, but requires "cost-scaled" inference during training which is intractable for many classes of output structures. However, we can use our notion of inference networks to circumvent this tractability issue and approximately optimize the slack-rescaled hinge loss, yielding the following optimization problem:

$$\min_\Theta \max_\Phi \sum_{\langle \boldsymbol{x}_i, \boldsymbol{y}_i \rangle \in \mathcal{D}} \triangle(\mathbf{A}_\Phi(\boldsymbol{x}_i), \boldsymbol{y}_i)[1 - E_\Theta(\boldsymbol{x}_i, \mathbf{A}_\Phi(\boldsymbol{x}_i)) + E_\Theta(\boldsymbol{x}_i, \boldsymbol{y}_i)]_+ \qquad (10)$$

Using the same argument as above, we can also break this into alternating optimization of $\Phi$ and $\Theta$.

We can optimize a structured perceptron (Collins, 2002) version by using the margin-rescaled hinge loss (Eq. (7)) and fixing $\triangle(\mathbf{A}_\Phi(\boldsymbol{x}_i), \boldsymbol{y}_i) = 0$. When using this loss, the cost-augmented inference network is actually a test-time inference network, because the cost is always zero, so using this loss may lessen the need to retune the inference network after training.

When we fix $\triangle(\mathbf{A}_\Phi(\boldsymbol{x}_i), \boldsymbol{y}_i) = 1$, then margin-rescaled hinge is equivalent to slack-rescaled hinge. While using $\triangle = 1$ is not useful in standard max-margin training with exact $\mathrm{argmax}$ inference (because the cost has no impact on optimization when fixed to a positive constant), it is potentially useful in our setting. Consider our SPEN objectives with $\triangle = 1$:

$$[1 - E_\Theta(\boldsymbol{x}_i, \mathbf{A}_\Phi(\boldsymbol{x}_i)) + E_\Theta(\boldsymbol{x}_i, \boldsymbol{y}_i)]_+ \qquad (11)$$

There will always be a nonzero difference between the two energies because $\mathbf{A}_\Phi(\boldsymbol{x}_i)$ will never exactly equal the discrete vector $\boldsymbol{y}_i$. Since there is no explicit minimization over all discrete vectors $\boldsymbol{y}$, this case is more similar to a "contrastive" hinge loss which seeks to make the energy of the true output lower than the energy of a particular "negative sample" by a margin of at least 1.

In our experiments, we will compare four hinge losses for training SPENs: margin-rescaled (Eq. (7)), slack-rescaled (Eq. (10)), perceptron (margin-rescaled with $\triangle = 0$), and contrastive ($\triangle = 1$).

## 5 IMPROVING TRAINING FOR INFERENCE NETWORKS

We found that the alternating nature of the optimization led to difficulties during training. Similar observations have been noted about other alternative optimization settings, especially those underlying generative adversarial networks (Salimans et al., 2016). Below we describe several techniques we found to help stabilize training, which are optional terms added to the objective in Eq. (8).

$L_2$ **Regularization:** We use $L_2$ regularization, adding the penalty term $\|\Phi\|_2^2$ with coefficient $\lambda_1$.

**Entropy Regularization:** We add an entropy-based regularizer $\mathrm{loss}_\mathrm{H}(\mathbf{A}_\Phi(\boldsymbol{x}))$ defined for the problem under consideration. For MLC, the output of $\mathbf{A}_\Phi(\boldsymbol{x})$ is a vector of scalars in $[0, 1]$, one for each label, where the scalar is interpreted as a label probability. The entropy regularizer $\mathrm{loss}_\mathrm{H}$ is the sum of the entropies over these label binary distributions. For sequence labeling, where the length of $\boldsymbol{x}$ is $N$ and where there are $L$ unique labels, the output of $\mathbf{A}_\Phi(\boldsymbol{x})$ is a length-$N$ sequence of length-$L$ vectors, each of which represents the distribution over the $L$ labels at that position in $\boldsymbol{x}$. Then, $\mathrm{loss}_\mathrm{H}$ is the sum of entropies of these label distributions across positions in the sequence.

When tuning the coefficient $\lambda_2$ for this regularizer, we consider both positive and negative values, permitting us to favor either low- or high-entropy distributions as the task prefers.[1]

**Local Cross Entropy Loss:** We add a local (non-structured) cross entropy $\mathrm{loss}_\mathrm{CE}(\mathbf{A}_\Phi(\boldsymbol{x}_i), \boldsymbol{y}_i)$ defined for the problem under consideration. We only experiment with this loss for sequence labeling.

---

[1]For MLC, encouraging lower entropy distributions worked better, while for sequence labeling, higher entropy was better, similar to the effect found by Pereyra et al. (2017). Further research is required to gain understanding of the role of entropy regularization in such alternating optimization settings.

It is the sum of the label cross entropy losses over all positions in the sequence. This loss provides more explicit feedback to the inference network, helping the optimization procedure to find a solution that minimizes the energy function while also correctly classifying individual labels. It can also be viewed as a multi-task loss for the inference network.

**Regularization Toward Pretrained Inference Network:** We add the penalty $\|\Phi - \Phi_0\|_2^2$ where $\Phi_0$ is a pretrained network, e.g., a local classifier trained to independently predict each part of $\boldsymbol{y}$.

Each additional term has its own tunable hyperparameter. Finally we obtain:

$$\hat{\Phi} \leftarrow \operatorname*{argmax}_{\Phi} \ [\triangle(\mathbf{A}_{\Phi}(\boldsymbol{x}_i), \boldsymbol{y}_i) - E_{\Theta}(\boldsymbol{x}_i, \mathbf{A}_{\Phi}(\boldsymbol{x}_i)) + E_{\Theta}(\boldsymbol{x}_i, \boldsymbol{y}_i)]_+ - \lambda_1 \|\Phi\|_2^2$$
$$+ \lambda_2 \text{loss}_{\text{H}}(\mathbf{A}_{\Phi}(\boldsymbol{x}_i)) - \lambda_3 \text{loss}_{\text{CE}}(\mathbf{A}_{\Phi}(\boldsymbol{x}_i), \boldsymbol{y}_i) - \lambda_4 \|\Phi - \Phi_0\|_2^2$$

## 6 RELATED WORK

Our methods are reminiscent of other alternating optimization problems like that underlying generative adversarial networks (GANs; Goodfellow et al. 2014). GANs are based on a minimax game and have a value function that one agent (a discriminator $D$) seeks to maximize and another (a generator $G$) seeks to minimize. By their analysis, a log loss discriminator converges to a degenerate uniform solution. When using hinge loss, we can get a non-degenerate discriminator while matching the data distribution (Dai et al., 2017; Zhao et al., 2016). Our formulation is closer to this hinge loss version of the GAN.

Our approach is also related to knowledge distillation (Ba & Caruana, 2014; Hinton et al., 2015), which refers to strategies in which one model (a "student") is trained to mimic another (a "teacher"). Typically, the teacher is a larger, more accurate model but which is too computationally expensive to use at test time. Urban et al. (2016) train shallow networks using image classification data labeled by an ensemble of deep teacher nets. Geras et al. (2016) train a convolutional network to mimic an LSTM for speech recognition. Others have explored knowledge distillation for sequence-to-sequence learning (Kim & Rush, 2016) and parsing (Kuncoro et al., 2016).

Since we train a single inference network for an entire dataset, our approach is also related to "amortized inference" (Srikumar et al., 2012; Gershman & Goodman, 2014; Paige & Wood, 2016; Chang et al., 2015). Such methods precompute or save solutions to subproblems for faster overall computation. Our inference networks likely devote more modeling capacity to the most frequent substructures in the data. A kind of inference network is used in variational autoencoders (Kingma & Welling, 2013) to approximate posterior inference in generative models.

Our methods are also related to work in structured prediction that seeks to approximate structured models with factorized ones, e.g., mean-field approximations in graphical models (Koller & Friedman, 2009; Krähenbühl & Koltun, 2011). Like our use of inference networks, there have been efforts in designing differentiable approximations of combinatorial search procedures (Martins & Kreutzer, 2017; Goyal et al., 2018) and structured losses for training with them (Wiseman & Rush, 2016). Since we relax discrete output variables to be continuous, there is also a connection to recent work that focuses on structured prediction with continuous valued output variables (Wang et al., 2016). They also propose a formulation that yields an alternating optimization problem, but it is based on proximal methods.

There are other settings in which gradient descent is used for inference, e.g., image generation applications like DeepDream (Mordvintsev et al., 2015) and neural style transfer (Gatys et al., 2015), as well as machine translation (Hoang et al., 2017). In these and related settings, gradient descent has started to be replaced by inference networks, especially for image transformation tasks (Johnson et al., 2016; Li & Wand, 2016). Our results below provide more evidence for making this transition. An alternative to what we pursue here would be to obtain an easier convex optimization problem for inference via input convex neural networks (Amos et al., 2017).

Table 1: Test F1 when comparing methods on multi-label classification datasets.

|  | Bibtex | Bookmarks | Delicious | avg. |
|---|---|---|---|---|
| MLP | 38.9 | 33.8 | 37.8 | 36.8 |
| SPEN (BM16) | **42.2** | 34.4 | **37.5** | 38.0 |
| SPEN (E2E) | 38.1 | 33.9 | 34.4 | 35.5 |
| SPEN (InfNet) | **42.2** | **37.6** | **37.5** | **39.1** |

## 7 EXPERIMENTS

In Sec. 7.1 we compare our approach to previous work on training SPENs for MLC. We compare accuracy and speed, finding our approach to outperform prior work. We then perform experiments with sequence labeling tasks in Sec. 7.2.

### 7.1 MULTI-LABEL CLASSIFICATION

We use the MLC datasets used by Belanger & McCallum (2016): Bibtex, Delicious, and Bookmarks. Dataset statistics are shown in Table 7 in the Appendix. For Bibtex and Delicious, we follow Belanger and McCallum and tune the hyperparameters using a different sampling of train and test data, then use the standard train/test split for final experimentation using the tuned hyperparameters. For Bookmarks, we use the same train/dev/test split as (Belanger & McCallum, 2016). For evaluation, we report the example averaged (macro averaged) F1 measure.

We use the SPEN for MLC described in Section 2 and also used by Belanger & McCallum (2016). For the feature representation network $F(\boldsymbol{x})$, we use feed-forward networks with two hidden layers, using their same layer widths: 150 for Bibtex/Bookmarks and 250 for Delicious. We pretrain the feature networks $F(\boldsymbol{x})$ by minimizing independent-label cross entropy for 10 epochs using Adam (Kingma & Ba, 2014) with learning rate 0.001. While training SPENs, we only update the parameters of the energy function ($\Theta$) and the inference network ($\Phi$), keeping the feature network parameters $F(\boldsymbol{x})$ fixed. We use Adam with learning rate 0.001 to train $\Theta$ and $\Phi$.

The inference networks are feed-forward networks with two hidden layers, using the same architectures as the feature networks $F(\boldsymbol{x})$. This permits us to initialize inference network parameters $\Phi$ using pretrained feature network parameters. For the output, we use an affine transformation layer with a sigmoid nonlinearity function, so the output values are in the range $(0, 1)$. We interpret each value as the probability of predicting the corresponding label. We obtain discrete predictions by thresholding at a threshold $\tau$ tuned to maximize F1 on the development data. We add three terms to the inference network objective from Section 5: $L_2$ regularization, entropy regularization, and regularization toward the pretrained feature network. Margin rescaling and slack rescaling use squared $L_2$ distance for $\triangle$. Additional details are provided in Sec. 9.1 in the appendix.

**Comparison to Prior Work.** Table 1 shows results comparing to prior work. The MLP and "SPEN (BM16)" baseline results are taken from (Belanger & McCallum, 2016). We obtained the "SPEN (E2E)" (Belanger et al., 2017) results by running the code available from the authors on these datasets. This method constructs a recurrent neural network that performs gradient-based minimization of the energy with respect to $\boldsymbol{y}$. They noted in their software release that, while this method is more stable, it is prone to overfitting and actually performs worse than the original SPEN. We indeed find this to be the case, as SPEN (E2E) underperforms SPEN (BM16) on all three datasets.

Our method ("SPEN (InfNet)") achieves the best average performance across the three datasets. It performs especially well on Bookmarks, which is the largest of the three. Our results use the contrastive hinge loss and retune the inference network on the development data after the energy is trained; these decisions were made based on the tuning described in Sec. 9.1, but all four hinge losses led to similarly strong results.

**Speed Comparison.** Table 2 compares training and test-time inference speed among the different methods. We only report speeds of methods that we ran.[2] The SPEN (E2E) times were obtained

---

[2] The MLP F1 scores above were taken from Belanger & McCallum (2016), but the MLP timing results reported in Table 2 are from our own experimental replication of their results.

Table 2: Training and test-time inference speed comparison (examples/sec).

| | Training Speed (examples/sec) | | | Testing Speed (examples/sec) | | |
|---|---|---|---|---|---|---|
| | Bibtex | Bookmarks | Delicious | Bibtex | Bookmarks | Delicious |
| MLP | 21670 | 19591 | 26158 | 90706 | 92307 | 113750 |
| SPEN (E2E) | 551 | 559 | 383 | 1420 | 1401 | 832 |
| SPEN (InfNet) | 5533 | 5467 | 4667 | 94194 | 88888 | 112148 |

using code obtained from Belanger and McCallum. We suspect that SPEN (BM16) training would be comparable to or slower than SPEN (E2E). Our method can process examples during training about 10 times as fast as the end-to-end SPEN, and 60-130 times as fast during test-time inference. In fact, at test time, our method is roughly the same speed as the MLP baseline, since our inference networks use the same architecture as the feature networks which form the MLP baseline. Compared to the MLP, the training of our method takes significantly more time overall because of joint training of the energy function and inference network, but fortunately the test-time inference is comparable.

## 7.2 SEQUENCE LABELING

We also evaluate our methods on sequence labeling. We report experiments with Twitter part-of-speech (POS) tagging here. Named entity recognition experiments are reported in the Appendix.

### 7.2.1 ENERGY FUNCTIONS FOR SEQUENCE LABELING

The input space $\mathcal{X}$ is now the set of all sequences of symbols drawn from a vocabulary. For an input sequence $\boldsymbol{x}$ of length $N$, where there are $L$ possible output labels for each position in $\boldsymbol{x}$, the output space $\mathcal{Y}(\boldsymbol{x})$ is $[L]^N$, where the notation $[q]$ represents the set containing the first $q$ positive integers. We define $\boldsymbol{y} = \langle y_1, y_2, .., y_N \rangle$ where each $y_i$ ranges over possible output labels, i.e., $y_i \in [L]$.

When defining our energy for sequence labeling, we take inspiration from bidirectional LSTMs (BLSTMs; Hochreiter & Schmidhuber 1997) and conditional random fields (CRFs; Lafferty et al. 2001). A "linear chain" CRF uses two types of features: one capturing the connection between an output label and $\boldsymbol{x}$ and the other capturing the dependence between neighboring output labels. We use a BLSTM to compute feature representations for $\boldsymbol{x}$. We use $f(\boldsymbol{x}, t) \in \mathbb{R}^d$ to denote the "input feature vector" for position $t$, defining it to be the $d$-dimensional BLSTM hidden vector at $t$.

We then define the following energy function:

$$E_\Theta(\boldsymbol{x}, \boldsymbol{y}) = -\left( \sum_t U_{y_t}^\top f(\boldsymbol{x}, t) + \sum_t W_{y_{t-1}, y_t} \right) \tag{12}$$

where $U_i \in \mathbb{R}^d$ is a parameter vector for label $i$ and the parameter matrix $W \in \mathbb{R}^{L \times L}$ contains label pair parameters. The full set of parameters $\Theta$ includes the $U_i$ vectors, $W$, and the parameters of the BLSTM. The above energy only permits discrete $\boldsymbol{y}$. For the general case that permits relaxing $\boldsymbol{y}$ to be continuous, we treat each $y_t$ as a vector. It will be one-hot for the ground truth $\boldsymbol{y}$ and will be a vector of label probabilities for relaxed $\boldsymbol{y}$'s. Then the general energy function is:

$$E_\Theta(\boldsymbol{x}, \boldsymbol{y}) = -\left( \sum_t \sum_{i=1}^L y_{t,i} \left( U_i^\top f(\boldsymbol{x}, t) \right) + \sum_t y_{t-1}^\top W y_t \right) \tag{13}$$

where $y_{t,i}$ is the $i$th entry of the vector $y_t$. In the discrete case, this entry is 1 for a single $i$ and 0 for all others, so this energy reduces to Eq. (12) in that case. In the continuous case, this scalar indicates the probability of the $t$th position being labeled with label $i$. For the label pair terms in this general energy function, we use a bilinear product between the vectors $y_{t-1}$ and $y_t$ using parameter matrix $W$, which also reduces to Eq. (12) when they are one-hot vectors.

**Tag Language Model.** In order to capture long-distance dependencies in an entire sequence of labels, we train a "tag language model" on a large corpus of automatically-tagged tweets, then include a term in the energy function representing the log-probability of the given tag sequence under this tag language model. Details are provided below in Section 7.2.4.

Table 3: Comparison of SPEN hinge losses and showing the impact of retuning (Twitter POS validation accuracies). Inference networks are trained with the cross entropy term.

| | validation accuracy (%) | |
|---|---|---|
| SPEN hinge loss | -retuning | +retuning |
| margin rescaling | 89.1 | 89.3 |
| slack rescaling | 89.4 | 89.6 |
| perceptron (MR, $\triangle = 0$) | 89.2 | 89.4 |
| contrastive ($\triangle = 1$) | 88.8 | 89.0 |

Table 4: Twitter POS accuracies of BLSTM, CRF, and SPEN (InfNet), using our tuned SPEN configuration (slack-rescaled hinge, inference network trained with cross entropy term). Though slowest to train, the SPEN matches the test-time speed of the BLSTM while achieving the highest accuracies.

| | validation accuracy (%) | test accuracy (%) | training speed (examples/sec) | testing speed (examples/sec) |
|---|---|---|---|---|
| BLSTM | 88.6 | 88.8 | 385 | 1250 |
| CRF | 89.1 | 89.2 | 250 | 500 |
| SPEN (InfNet) | 89.6 | 89.8 | 125 | 1250 |

### 7.2.2 EXPERIMENTAL SETUP

For Twitter part-of-speech (POS) tagging, we use the annotated data from Gimpel et al. (2011) and Owoputi et al. (2013) which contains $L = 25$ POS tags. For training, we combine the 1000-tweet OCT27TRAIN set and the 327-tweet OCT27DEV set. For validation, we use the 500-tweet OCT27TEST set and for testing we use the 547-tweet DAILY547 test set. We use 100-dimensional skip-gram embeddings trained on 56 million English tweets with `word2vec` (Mikolov et al., 2013).[3]

We use a BLSTM to compute the "input feature vector" $f(\boldsymbol{x}, t)$ for each position $t$, using hidden vectors of dimensionality $d = 100$. We also use BLSTMs for the inference networks. The output layer of the inference network is a softmax function, so at every position, the inference network produces a distribution over labels at that position. We train inference networks using stochastic gradient descent (SGD) with momentum and train the energy parameters using Adam. For $\triangle$, we use $L_1$ distance. We tune hyperparameters on the validation set; full details of tuning are provided in the appendix. We found that the cross entropy stabilization term worked well for this setting; details and an empirical comparison are provided in Section 9.2.1.

We compare to standard BLSTM and CRF baselines. We train the BLSTM baseline to minimize per-token log loss; this is often called a "BLSTM tagger". We train a CRF baseline using the energy in Eq. (12) with the standard conditional log-likelihood objective using the standard dynamic programming algorithms (forward-backward) to compute gradients during training. Further details are provided in the appendix.

### 7.2.3 RESULTS

**Loss Function Comparison.** Table 3 shows results when comparing SPEN training objectives. We see a larger difference among losses here than for MLC tasks. When using the perceptron loss, there is no margin, which leads to overfitting: 89.4 on validation, 88.6 on test (not shown in the table). The contrastive loss, which strives to achieve a margin of 1, does better on test (89.0). We also see here that margin rescaling and slack rescaling both outperform the contrastive hinge, unlike the MLC tasks. We suspect that in the case in which each input/output has a different length, using a cost that captures length is more important.

**Comparison to Standard Baselines.** Table 4 compares our final tuned SPEN configuration to two standard baselines: a BLSTM tagger and a CRF. The SPEN achieves higher validation and test accuracies with faster test-time inference. While our method is slower than the baselines during

---

[3]The pretrained embeddings are the same as those used by Tu et al. (2017) and are available at `http://ttic.uchicago.edu/~lifu/`

Table 5: Twitter POS validation/test accuracies when adding tag language model (TLM) energy term to a SPEN trained with margin-rescaled hinge.

|       | val. accuracy (%) | test accuracy (%) |
|-------|-------------------|-------------------|
| -TLM  | 89.8              | 89.6              |
| +TLM  | 89.9              | 90.2              |

training, it is faster than the CRF at test time, operating at essentially the same speed as the BLSTM baseline while being more accurate.

Here, the SPEN and CRF are using the same functional form for their energy functions, namely the energy given in Eq. (13). We note that the SPEN outperforms the CRF, despite using the same form for the energy. There are two factors that can explain this. First, the losses are different. The CRF uses conditional log-likelihood while the SPEN results here use slack-rescaled hinge, which outperforms the other hinge loss variants (Table 3). Second, the stabilization terms used when training the inference network may be providing a regularizing effect for the model. Our motivation for these experiments was to show the impact of these differences while keeping the form of the energy function fixed. We now turn to richer energies.

### 7.2.4 TOWARDS GLOBAL ENERGIES: TAG LANGUAGE MODELS FOR TWITTER POS TAGGING

The above results only use the pairwise energy; no results used the tag language model (TLM). To compute the TLM energy term, we first automatically tag unlabeled tweets, then train an LSTM language model on the automatic tag sequences. When doing so, we define the input tag embeddings to be $L$-dimensional one-hot vectors specifying the tags in the training sequences. This is nonstandard compared to standard language modeling. In standard language modeling, we train on observed sequences and compute likelihoods of other fully-observed sequences. However, in our case, we train on tag sequences but we want to use the same model on sequences of tag *distributions* produced by an inference network. We train the TLM on sequences of one-hot vectors and then use it to compute likelihoods of sequences of tag distributions. Further details about training are provided in Section 9.2.2 in the appendix.

We define an additional energy term $E^{\mathrm{TLM}}(\boldsymbol{y})$ based on the pretrained TLM. If the argument $\boldsymbol{y}$ consisted of one-hot vectors, we could simply compute its likelihood. However, to support relaxed $\boldsymbol{y}$'s, we need to define a more general function:

$$E^{\mathrm{TLM}}(\boldsymbol{y}) = -\sum_{t=1}^{|\boldsymbol{y}|+1} \log(y_t^\top \mathrm{TLM}(\langle y_0, ..., y_{t-1} \rangle)) \tag{14}$$

where $y_0$ is the start-of-sequence symbol, $y_{|\boldsymbol{y}|+1}$ is the end-of-sequence symbol, and $\mathrm{TLM}(\langle y_0, ..., y_{t-1} \rangle)$ returns the softmax distribution over tags at position $t$ (under the pretrained tag language model) given the preceding tag vectors. When each $y_t$ is a one-hot vector, this energy reduces to the negative log-likelihood of the tag sequence specified by $\boldsymbol{y}$.

We define the new joint energy as the sum of the energy function in Eq. (13) and the TLM energy function in Eq. (14). During learning, we keep the TLM parameters fixed to their pretrained values, but we tune the weight of the TLM energy (over the set $\{0.1, 0.2, 0.5\}$) in the joint energy. We train SPENs with the new joint energy using the margin-rescaled hinge, training the inference network with the cross entropy term.

Table 5 shows results.[4] Adding the TLM energy leads to a gain of 0.6 on the test set. Other settings showed more variance; when using slack-rescaled hinge, we found a small drop on test, while when simply training inference networks for a fixed, pretrained joint energy with tuned mixture coefficient, we found a gain of 0.3 on test when adding the TLM energy. We investigated the improvements and found some to involve corrections that seemingly stem from handling non-local dependencies better. Table 10 in the appendix shows examples in which the model with the TLM appears to be better at using the broader context when making tagging decisions. These results suggest that our method of

---

[4] The baseline results differ slightly from earlier results because we found that we could achieve higher accuracies in SPEN training by avoiding using pretrained feature network parameters for the inference network.

Table 6: Comparison of test-time inference algorithms for a trained CRF (Twitter POS tagging). We show the test accuracy for the inference network setting that does best on validation. All inference networks use the same architecture and therefore have essentially the same speed.

| test-time inference algorithm | val. accuracy (%) | test accuracy (%) | speed (examples/sec) |
|---|---|---|---|
| Viterbi algorithm | 89.1 | 89.2 | 500 |
| Inference network + cross entropy | 89.7 | 89.5 | 1250 |
| Inference network + entropy | 89.6 | | |
| Inference network + squared L2 distance | 88.9 | | |

training inference networks can be used to add rich features to structured prediction, though we leave a thorough exploration of global energies to future work.

### 7.2.5 BEYOND SPENS: INFERENCE NETWORKS FOR STRUCTURED PREDICTION

We note that inference networks can be used for any prediction problem. We now explore the use of an inference network to approximate test-time inference for a trained CRF. The results are shown in Table 6. All results use the same trained CRF energy function (Eq. (12)), trained to minimize log loss using the forward-backward algorithm for exact inference during training. The first row shows accuracy and speed when using Viterbi for test-time inference, which is the same setting as the "CRF" row in Table 4. Subsequent rows show results when training inference networks to mimic Viterbi with various stabilization terms. When training these inference networks, we train them on the training set and tune based on early stopping on the validation set. The energy stays fixed while inference networks are trained.

When using either entropy or cross entropy, our inference networks outperform Viterbi while doubling its speed. When using the squared $L_2$ distance term (which regularizes the inference network toward the pretrained BLSTM), the accuracy reduces to be closer to that of the BLSTM, which reaches 88.6% on validation (see Table 4). When using no stabilization terms for the inference network, learning fails, reaching 13.7% on the development set, showing the importance of using some stabilization term while training the inference network.

These results show promise for training inference networks to speed up combinatorial algorithms for structured prediction and other domains.

## 8 CONCLUSIONS AND FUTURE WORK

We presented ways to jointly train structured energy functions and inference networks using large-margin objectives. The energy function captures arbitrary dependencies among the labels, while the inference networks learns to capture the properties of the energy in an efficient manner, yielding fast test-time inference. Future work includes exploring the space of network architectures for inference networks to balance accuracy and efficiency, experimenting with additional global terms in structured energy functions, and exploring richer structured output spaces such as trees and sentences.

### ACKNOWLEDGMENTS

We thank the anonymous reviewers, David Belanger, Weiran Wang and Zheng Cai. We also thank NVIDIA Corporation for donating GPUs used in this research.

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

Table 7: Statistics of the multi-label classification datasets.

| | # labels | # features | # train | # dev | # test |
|---|---|---|---|---|---|
| Bibtex | 159 | 1836 | 4836 | - | 2515 |
| Bookmarks | 208 | 2151 | 48000 | 12000 | 27856 |
| Delicious | 982 | 501 | 12896 | - | 3185 |

Table 8: Development F1 for Bookmarks when comparing hinge losses for SPEN (InfNet) and whether to retune the inference network.

| hinge loss | -retuning | +retuning |
|---|---|---|
| margin rescaling | 38.51 | 38.68 |
| slack rescaling | 38.57 | 38.62 |
| perceptron (MR, $\triangle = 0$) | 38.55 | 38.70 |
| contrastive ($\triangle = 1$) | 38.80 | 38.88 |

## 9 APPENDIX

### 9.1 MULTI-LABEL CLASSIFICATION

Table 7 shows dataset statistics for the multi-label classification datasets.

**Hyperparameter Tuning.** We tune $\lambda$ (the $L_2$ regularization strength for $\Theta$) over the set $\{0.01, 0.001, 0.0001\}$. The classification threshold $\tau$ is chosen from $[0, 0.01, 0.02, 0.03, 0.04, 0.05, 0.1, 0.15, 0.2, 0.25, 0.3, 0.35, 0.4, 0.45, 0.5, 0.55, 0.6, 0.65, 0.7, 0.75]$ as also done by Belanger & McCallum (2016). We tune the coefficients for the three stabilization terms for the inference network objective from Section 5 over the follow ranges: $L_2$ regularization ($\lambda_1 \in \{0.01, 0.001, 0.0001\}$), entropy regularization ($\lambda_2 = 1$), and regularization toward the pretrained feature network ($\lambda_4 \in \{0, 1, 10\}$).

**Comparison of Loss Functions and Impact of Inference Network Retuning.** Table 8 shows results comparing the four loss functions from Section 4.2 on the development set for Bookmarks, the largest of the three datasets. We find performance to be highly similar across the losses, with the contrastive loss appearing slightly better than the others.

After training, we "retune" the inference network as specified by Eq. (5) on the development set for 20 epochs using a smaller learning rate of 0.00001. Table 8 shows slightly higher F1 for all losses with retuning. We were surprised to see that the final cost-augmented inference network performs well as a test-time inference network. This suggests that by the end of training, the cost-augmented network may be approaching the argmin and that there may not be much need for retuning.

When using $\triangle = 0$ or 1, retuning leads to the same small gain as when using the margin-rescaled or slack-rescaled losses. Here the gain is presumably from adjusting the inference network for other inputs rather than from converting it from a cost-augmented to a test-time inference network.

### 9.2 TWITTER POS TAGGING

#### 9.2.1 HYPERPARAMETER TUNING

When training inference networks and SPENs for Twitter POS tagging, we use the following hyperparameter tuning. We tune the inference network learning rate ($\{0.1, 0.05, 0.02, 0.01, 0.005, 0.001\}$), $L_2$ regularization ($\lambda_1 \in \{0, 1e-3, 1e-4, 1e-5, 1e-6, 1e-7\}$), the entropy regularization term ($\lambda_2 \in \{0.1, 0.5, 1, 2, 5, 10\}$), the cross entropy regularization term ($\lambda_3 \in \{0.1, 0.5, 1, 2, 5, 10\}$), and the squared L2 distance ($\lambda_4 \in \{0, 0.1, 0.2, 0.5, 1, 2, 10\}$). We train the energy functions with Adam with a learning rate of 0.001 and $L_2$ regularization ($\lambda_1 \in \{0, 1e-3, 1e-4, 1e-5, 1e-6, 1e-7\}$).

Table 9 compares the use of the cross entropy and entropy stabilization terms when training inference networks for a SPEN with margin-rescaled hinge. Cross entropy works better than entropy in this setting, though retuning permits the latter to bridge the gap more than halfway.

Table 9: Comparison of inference network stabilization terms and showing impact of retuning when training SPENs with margin-rescaled hinge (Twitter POS validation accuracies).

|  | validation accuracy (%) | |
| --- | --- | --- |
| inference network stabilization terms | -retuning | +retuning |
| cross entropy | 89.1 | 89.3 |
| entropy | 84.2 | 86.8 |

Table 10: Examples of improvements in Twitter POS tagging when using tag language model (TLM). In all of these examples, the predicted tag when using the TLM matches the gold standard.

|  |  | predicted tags | |
| --- | --- | --- | --- |
| # | tweet (target word in bold) | -TLM | +TLM |
| 1 | ... that's a t-17 , technically . does **that** count as top-25 ? | determiner | pronoun |
| 2 | ... lol you know im down **like** 4 flats on a cadillac ... lol ... | adjective | preposition |
| 3 | ... them who he is : he wants her to **like** him for his pers ... | preposition | verb |
| 4 | I wonder when Nic Cage is going to **film** " Another Something Something Las Vegas " . | noun | verb |
| 5 | Cut my hair , **gag** and bore me | noun | verb |
| 6 | ... they had their fun , we **hd** ours ! ;) lmaooo | proper noun | verb |
| 7 | " Logic will get you from A to **B** . Imagination will take you everywhere . " - Albert Einstein . | verb | noun |
| 8 | lmao I'm not a sheep who listens to it **cos** everyone else does ... | verb | preposition |
| 9 | Noo its not cuss you have swag **andd** you wont look dumb ! ... | noun | coord. conj. |

When training CRFs, we use SGD with momentum. We tune the learning rate (over $\{0.1, 0.05, 0.02, 0.01, 0.005, 0.001\}$) and $L_2$ regularization coefficient (over $\{0, 1e-3, 1e-4, 1e-5, 1e-6, 1e-7\}$). For all methods, we use early stopping based on validation accuracy.

### 9.2.2 TAG LANGUAGE MODEL DETAILS AND ANALYSIS

To obtain training data for training the tag language model, we run the Twitter POS tagger from Owoputi et al. (2013) on a dataset of 303K randomly-sampled English tweets. We train the tag language model on 300K tweets and use the remaining 3K for tuning hyperparameters and early stopping. We train an LSTM language model on the tag sequences using stochastic gradient descent with momentum and early stopping on the validation set. We used a dropout rate of 0.5 for the LSTM hidden layer. We tune the learning rate ($\{0.1, 0.2, 0.5, 1.0\}$), the number of LSTM layers ($\{1, 2\}$), and the hidden layer size ($\{50, 100, 200\}$).

Table 10 shows examples in which our SPEN that includes the TLM appears to be using broader context when making tagging decisions. These are examples from the test set labeled by two models: the SPEN without the TLM (which achieves 89.6% accuracy, as shown in Table 5) and the SPEN with the TLM (which reaches 90.2% accuracy). In example 1, the token "that" is predicted to be a determiner based on local context, but is correctly labeled a pronoun when using the TLM. This example is difficult because of the noun/verb tag ambiguity of the next word ("count") and its impact on the tag for "that". Examples 2 and 3 show two corrections for the token "like", which is a highly ambiguous word in Twitter POS tagging. The broader context makes it much clearer which tag is intended.

The next two examples (4 and 5) are cases of noun/verb ambiguity that are resolvable with larger context. The last four examples show improvements for nonstandard word forms. The shortened form of "had" (example 6) is difficult to tag due to its collision with "HD" (high-definition), but the model with the TLM is able to tag it correctly. In example 7, the ambiguous token "b" is frequently used as a short form of "be" on Twitter, and since it comes after "to" in this context, the verb interpretation is encouraged. However, the broader context makes it clear that it is not a verb and the TLM-enriched model tags it correctly. The words in the last two examples are nonstandard word forms that were not observed in the training data, which is likely the reason for their erroneous predictions. When using the TLM, we can better handle these rare forms based on the broader context.

Table 11: Named entity recognition F1 of BLSTM, CRF, and SPEN (InfNet) with slack-rescaled hinge where inference networks used cross entropy stabilization term. Though slowest to train, the SPEN matches the test-time speed of the BLSTM while improving F1 by 2 points, though it lags behind the CRF.

|  | validation F1 | test F1 | training speed (examples/sec) | testing speed (examples/sec) |
|---|---|---|---|---|
| BLSTM | 88.30 | 83.02 | 385 | 1042 |
| CRF | 91.31 | 87.15 | 222 | 454 |
| SPEN (InfNet) | 89.98 | 85.06 | 118 | 1025 |

### 9.2.3 LEARNED PAIRWISE POTENTIAL MATRIX

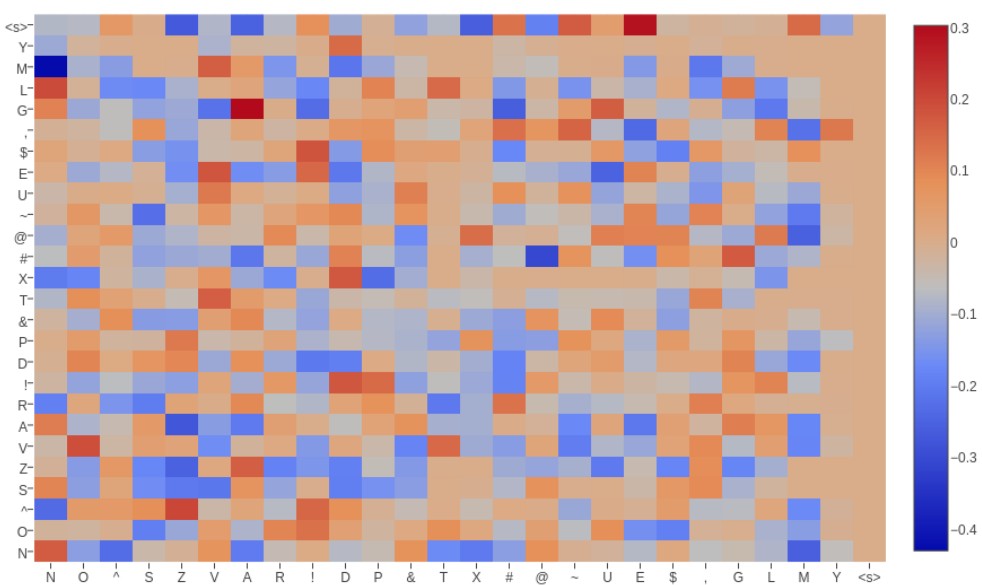

Figure 1: Learned pairwise potential matrix for Twitter POS tagging.

Figure 1 shows the learned pairwise potential matrix $W$ in Twitter POS tagging. We can see strong correlations between labels in neighborhoods. For example, an adjective (A) is more likely to be followed by a noun (N) than a verb (V) (see row labeled "A" in the figure).

### 9.3 NAMED ENTITY RECOGNITION

For named entity recognition (NER), we perform experiments on the English data from the CoNLL 2003 shared task (Tjong Kim Sang & De Meulder, 2003). This task contains sentences annotated with named entities and their types, containing 14987 training sentences, 3466 in the development set, and 3684 in the test set. There are four named entity types: PERSON, LOCATION, ORGANIZA-TION, and MISC. We use the BIOES tagging scheme instead of the original BIO2, following prior work (Ratinov & Roth, 2009; Ma & Hovy, 2016). There are $L = 17$ classes. We use 100-dimensional pretrained GloVe (Pennington et al., 2014) embeddings trained on 6 billion words from Wikipedia and web text, which work better than other pretrained embeddings (Ma & Hovy, 2016).

Results are shown in Table 11. We see a large 4-point gap between the BLSTM and CRF, suggesting the importance of structured information for this problem. Though the SPEN still lags behind the CRF in F1, it matches the test-time speed of the BLSTM while improving F1 by 2 points.

