# OpenReview forum: "Learning Approximate Inference Networks for Structured Prediction"
_ICLR.cc/2018/Conference — Accept (Poster)_

### Official Review · AnonReviewer3 · 2017-11-27
**Amortized inference for SPENs**

**Rating:** 7
**Confidence:** 5

**Review:**

= Quality =
Overall, the authors do a good job of placing their work in the context of related research, and employ a variety of non-trivial technical details to get their methods to work well.

= Clarity =

Overall, the exposition regarding the method is good. I found the setup for the sequence tagging experiments confusing, tough. See more comments below.

= Originality / Significance =

The paper presents a clever idea that could help make SPENs more practical. The paper's results also suggest that we should be thinking more broadly about how to using complicated structured distributions as teachers for model compression.

= Major Comment =

I'm concerned by the quality of your results and the overall setup of your experiments. In particular, the principal contribution of the sequence tagging experiments seems top be different than what is advertised earlier on in the paper.

Most of your empirical success is obtained by taking a pretrained CRF energy function and using this as a teacher model to train a feed-forward inference network. You have have very few experiments using a SPEN energy function parametrization that doesn't correspond to a CRF, even though you could have used an arbitrary convnet, RNN, etc. The one exception is when you use the tag language model. This is a good idea, but it is pretrained, not trained using the saddle-point objective you introduce. In fact, you don't have any results demonstrating that the saddle-point approach is better than simpler alternatives.

It seems that you could have written a very different paper about model compression with CRFs that would have been very interesting and you could've have used many of the same experiments. It's unclear why SPENs are so important. The idea of amortizing inference is perhaps more general. My recommendation is that you either rebrand the paper to be more about general methods for amortizing structured prediction inference using model compression or do more fine-grained experiments with SPENs that demonstrate empirical gains that leverage their flexible deep-network-based energy functions.


= Minor Comments =

* You should mention 'Energy Based GANs"

* I don't understand "This approach performs backpropagation through each step of gradient descent, permitting more stable training but also evidently more overfitting." Why would it overfit more? Simply because training was more stable? Couldn't you prevent overfitting by regularizing more?

* You spend too much space talking about specific hyperparameter ranges, etc. This should be moved to the appendix. You should also add a short summary of the TLM architecture to the main paper body.

* Regarding your footnote discussing using a positive vs. negative sign on the entropy regularization term, I recommend checking out "Regularizing neural networks by penalizing confident output distributions."

* You should add citations for the statement "In these and related settings, gradient descent has started to be replaced by inference networks."

* I didn't find Table 1 particularly illuminating. All of the approaches seem to perform about the same. What conclusions should I make from it?

* Why not use KL divergence as your \Delta function?

* Why are the results in Table 5 on the dev data?

* I was confused by Table 4. First of all, it took me a very long time to figure out that the middle block of results corresponds to taking a pretrained CRF energy and amortizing inference by training an inference network. This idea of training with a standard loss (conditional log lik.) and then amortizing inference post-hoc was not explicitly introduced as an alternative to the saddle point objective you put forth earlier in the paper. Second, I was very surprised that the inference network outperformed Viterbi (89.7 vs. 89.1 for the same CRF energy). Why is this?

* I'm confused by the difference between Table 6 and Table 4? Why not just include the TLM results in Table 4?

---

> ### Author Response · Authors · 2018-01-03
> **Response**
>
> Thanks very much for the thoughtful review!
>
> Regarding your major comment, we will first mention that the revised version includes additional experimental results when using our framework to train a SPEN with a global energy that includes the tag language model (TLM) energy. These results are described in Sec. 7.2.4.
>
> We agree that the original submission suffered from a bit of an identity crisis. As you mentioned, “The idea of amortizing inference is perhaps more general” and we intend to develop this direction in future work. Also, in the revised version, we restructured the sequence labeling section so as to more cleanly separate the discussion of training SPENs (Sec. 7.2.3) and exploring richer energy functions (Sec. 7.2.4) from the discussion of amortizing inference for pretrained structured predictors (Sec. 7.2.5).
>
> Replies to your minor comments are below:
>
> “Energy Based GANs”
> Thanks -- we added a mention and citation to the Related Work section.
>
> “Why would it overfit more? Simply because training was more stable? Couldn't you prevent overfitting by regularizing more?”
>
> We should have provided a citation for this. The github page hosting the SPEN code includes the claim: “the end-to-end approach fitting the training data much better is that it is more prone to overfitting”. As that method is from prior work, we do not know exactly what the cause is of the observed overfitting. It may be that it is caused by the increased capability of calculating precise gradients obtained by unrolling gradient descent into a computation graph, rather than merely performing gradient descent for inference in an offline manner. We clarified the above in Sec. 7.1.
>
> “Move hyperparameter ranges, etc to the appendix. Add summary of TLM architecture to the main paper”
>
> We moved several tuning details to the appendix and moved the TLM description to the main body (Sec. 7.2.4).
>
> “footnote on using positive vs. negative sign on entropy regularization term”
>
> Thanks for the pointer!  We added a citation.
>
> “add citations for ‘gradient descent has started to be replaced by inference networks.’”
>
> Good point. We added relevant citations to that claim.
>
> “Table 1 not particularly illuminating. All of the approaches seem to perform about the same.”
>
> Usually, using cost-augmented inference for testing (with an SVM) gives really bad predictions.  We were surprised to see that the final cost-augmented inference network performs well as a test-time inference network. This suggests that by the end of training, the cost-augmented network may be approaching the argmin. Nonetheless, since the differences are small there, we moved this table and discussion of this to the appendix.
>
> “Why are results in Table 5 on dev?”
> We often reported results only on dev so as to avoid reporting too many configurations on the test set, in order to prevent us (and the community) from learning too much about what works best on the test set.
>
> “Why not use KL divergence as your \Delta function?”
>
> In classic max-margin structured prediction, \Delta is a symmetric function, so we didn’t consider using KL divergence. But we could use JS divergence and we think an exploration of the choices here would be interesting future work. (Also, we could try using asymmetric \Delta functions as there does not appear to be any strong theoretical motivation to use symmetric \Delta functions (in our view); it appears to be mostly just a convention.)
>
> “confused by Table 4”
>
> Thanks to the comments by you and the other reviewers, we heavily modified Table 4, splitting it into multiple simpler tables (see the new tables 4, 6, and 9).
>
> “very surprised that the inference network outperformed Viterbi (89.7 vs. 89.1 for the same CRF energy). Why is this?”
>
> This is a good question.  We added some speculation to Sec. 7.2.3 that we think is relevant to this question as well.  In particular, the stabilization terms used when training the inference network may be providing a regularizing effect for the model.
>
> “confused by difference between Table 6 and Table 4”
>
> Yes, we agree that was confusing. We restructured both tables. Please see the new tables 4, 5, and 6.

---

> ### Comment · AnonReviewer3 · 2018-01-09
> **Response to the authors' comments and version 2 of the paper.**
>
> The new experiments sections is substantially better. It does a good job of providing separate analyses of the various contributions of the paper. Overall, there is definitely a wealth of follow-on work to be done in this area, and the ICLR community will appreciate this paper.

---

### Official Review · AnonReviewer2 · 2017-11-27
**New idea for training structured predictors, but unclear motivation and evaluation**

**Rating:** 5
**Confidence:** 3

**Review:**

The paper proposes training ``inference networks,'' which are neural network structured predictors. The setup is analogous to generative adversarial networks, where the role of the discriminator is played by a structured prediction energy network (SPEN) and the generator is played by an inference network.

The idea is interesting. It could be viewed as a type of adversarial training for large-margin structured predictors, where counterexamples, i.e., structures with high loss and low energy, cannot be found by direct optimization. However, it remains unclear why SPENs are the right choice for an energy function.

Experiments suggest that it can result in better structured predictors than training models directly via backpropagation gradient descent. However, the experimental results are not clearly presented. The clarity is poor enough that the paper might not be ready for publication.

Comments and questions:

1) It is unclear whether this paper is motivated by training SPENs or by training structured predictors. The setup focuses on using SPENs as an inference network, but this seems inessential. Experiments with simpler energy functions seem to be absent, though the experiments are unclear (see below).

2) The confusion over the motivation is confounded by the fact that the experiments are very unclear. Sometimes predictions are described as the output of SPENs (Tables 2, 3, 4, and 7), sometimes as inference networks (Table 5), and sometimes as a CRF (Tables 4 and 6). In 7.2.2 it says that a BiLSTM is used for the inference network in Twitter POS tagging, but Tables 4 and 6 indicate both CRFs and BiLSTMS? It is also unclear when a model, e.g., BiLSTM or CRF is the energy function (discriminator) or inference network (generator).

3) The third and fourth columns of Table 5 are identical. The presentation should be made consistent, either with dev/test or -retuning/+retuning as the top level headers.

4) It is also unclear how to compare Tables 4 and 5. The second to bottom row of Table 5 seems to correspond with the first row of Table 5, but other methods like slack rescaling have higher performance. What is the takeaway from these two tables supposed to be?

5) Part of the motivation for the work is said to be the increasing interest in inference networks: "In these and related settings, gradient descent has started to be replaced by inference networks. Our results below provide more evidence for making this transition." However, no other work on inference networks is directly cited.

---

> ### Author Response · Authors · 2018-01-03
> **Response**
>
> Thanks for the questions. We agree with you that the experiment description was unclear in many places and we think the revised version is much improved in this regard. Specific answers to your numbered questions are below:
>
> 1) The primary goal of the paper is to propose a framework to do training and inference with SPENs.  We have rewritten the experimental results section to focus on evaluating this SPEN training/inference framework.  It turns out that the framework can also be applied to simpler families of structured prediction models, so we also include experimental results for applying inference network training to CRFs (see Sec. 7.2.5 in the revised version).  In the new version, we have tried to more cleanly separate the contributions for SPENs from the contributions to structured prediction more generally, by relegating the latter results to Sec. 7.2.5 only.
>
> 2) All good points. We hope the revised version will help resolve all of these confusions. Please let us know if anything is still unclear.
>
> 3) We restructured this table to remove redundant columns and make the presentation simpler.
>
> 4) Thanks to the comments by you and the other reviewers, we heavily modified Table 4, splitting it into multiple simpler tables (see the new tables 4, 6, and 9).
>
> 5) Good point. We added relevant citations to that claim.

---

### Official Review · AnonReviewer1 · 2017-11-28
**Interesting reinterpretation of SPENs**

**Rating:** 9
**Confidence:** 4

**Review:**

This paper proposes an improvement in the speed of training/inference with structured prediction energy networks (SPENs) by replacing the inner optimization loop with a network trained to predict its outputs.

SPENs are an energy-based structured prediction method, where the final prediction is obtained by optimizing min_y E_theta(f_phi(x), y), i.e., finding the label set y with the least energy, as computed by the energy function E(), using a set of computed features f_phi(x) which comes from a neural network. The key innovation in SPENs was representing the energy function E() as an arbitrary neural network which takes the features f(x) and candidate labels y and outputs a value for the energy. At inference time y can be optimized by gradient descent steps. SPENs are trained using maximum-margin loss functions, so the final optimization problem is max -loss(y, y') where y' = argmin_y E(f(x), y).

The key idea of this paper is to replace the minimization of the energy function min_y E(f(x), y) with a neural network which is trained to predict the resulting output of this minimization. The resulting formulation is a min-max problem at training time with a striking similarity to the GAN min-max problem, where the y-predicting network learns to predict labels with low energy (according to the E-computing network) and high loss while the energy network learns to assign a high energy to predicted labels which have a higher loss than true labels (i.e. the y-predicting network acts as a generator and the E-predicting network acts as a discriminator).

The paper explores multiple loss functions and techniques to train these models. They seem rather finnicky, and the experimental results aren't particularly strong when it comes to improving the quality over SPENs but they have essentially the same test-time complexity as simple feedforward models while having accuracy comparable to full inference-requiring energy-based models. The improved understanding of SPENs and potential for further work justify accepting this paper.

---

> ### Author Response · Authors · 2018-01-03
> **Thanks**
>
> Thank you for the comments and the support!

---

### Author Response · Authors · 2018-01-03
**We posted a revised version of paper and add some comments.**

Thanks to the reviewers for the many comments and questions. We just posted a revised version that we think addresses many of them. In particular, we rewrote the sequence labeling experimental section (Sec. 7.2).  We simplified the experimental settings in each table to make the results easier to understand.  The results were admittedly very confusing, as we were combining different energy functions, training objectives, and inference network architectures, all in the same table.  We hope we have corrected that with the new rewrite.  We’ll give a quick summary of our changes below:

Table 4 compares our tuned SPEN configuration (which we are now calling “SPEN (InfNet)” throughout) to off-the-shelf BLSTM and CRF baselines.  The SPEN and CRF in that table use the same energy, namely the energy given in Eq. (13).  These experiments allow us to show the impact of differences in training and the use of inference networks while keeping the form of the energy function fixed.

But, as multiple reviewers pointed out, the goal of SPENs is to use energies that go beyond what’s possible with traditional models like chain CRFs.  We definitely agree with this.  While we intend to pursue this more thoroughly in future work, we do feel that the tag language model (TLM) results are a promising step in this direction.  In Sec. 7.2.4, we describe the tag language model energy and present results when adding it to the energy in Eq. (13) and training with our framework.

Then, in Sec. 7.2.5, we describe experiments in training inference networks to do test-time inference for a pretrained, off-the-shelf CRF.  These results were admittedly confusing in the  original submission, but hopefully by separating them out and moving them to the end of the paper, it is now more clear.  We agree with the reviewers that the approach described (of training inference networks to approximate prediction problems) does indeed apply beyond SPENs. While we did not have space to thoroughly explore this application in this submission, we hope that this small section of promising experimental results will help other researchers to see the potential of inference networks for structured prediction more broadly.

---

### Decision · Program_Chairs · 2018-01-29
**ICLR 2018 Conference Acceptance Decision**

**Decision:**

Accept (Poster)

**Comment:**

The submission modifies the SPEN framework for structured prediction by adding an inference network in place of the usual combinatorial optimization based inference.  The resulting architecture has some similarity to a GAN, and significantly increases the speed of inference.

The submission provides links between two seemingly different frameworks: SPENs and GANs.  By replacing inference with a network output, the connection is made, but importantly, this massively speeds up inference and may mark an important step forward in structured prediction with deep learning.